# Fostering Open Data Practices in Research-Performing Organizations

Claire Jean-Quartier [1,*], Harald Kleinberger-Pierer [2], Barbara Zach [3], Ilire Hasani-Mavriqi [1], Lea Pešec [3] and Clara Schmikl-Reiter [4]

1 Research Data Management, Graz University of Technology, 8010 Graz, Austria
2 Research Management and Service, University of Graz, 8010 Graz, Austria; harald.kleinberger-pierer@uni-graz.at
3 IDea_Lab, University of Graz, 8010 Graz, Austria; barbara.zach@uni-graz.at (B.Z.); lea.pesec@uni-graz.at (L.P.)
4 Research Organization and Services, FH JOANNEUM, 8020 Graz, Austria; clara.schmikl-reiter@fh-joanneum.at
* Correspondence: c.jean-quartier@tugraz.at

**Abstract:** Open data provide the scientific community and other stakeholders with unrestricted access to data. Open data serve as a foundation for reproducing research findings, while also facilitating collaboration and enabling novel discoveries. However, open data practices are still not commonly applied. To contribute to the implementation of open data strategy in academia in Austria and beyond, a collection of local strategies from regional universities and higher education institutions in the Austrian provinces of Styria and Carinthia was compiled through workshop-based discussions between participants from research support service units at research-performing organizations. The collection was further organized into categories based on application time scenarios, target groups, and involved parties, as well as corresponding thematic focus. A strategic guide consisting of various measures has been developed to encourage the adoption of open data practices from an organizational standpoint. Designed for adaptability, it aims to be applicable and modifiable by all interested research and higher education institutions, regardless of their priorities and resources. Our guideline aids research organizations in crafting a tailored strategy to enhance their data dissemination practices, thereby increasing their research visibility and impact.

**Keywords:** open data; data management; research organization; knowledge transfer; public access; higher education institution



## 1. Introduction

Open data refer to data freely available for reuse without restrictions. The term was initially coined within the scientific community [1]. It extends beyond science into public domains like the economy, society, and governance. The latter even promotes open data more vigorously than many institutions of higher education themselves [2,3]. Open data constitute an integral part of open science, enhancing reusability and enabling validation through reproducibility and transparency [4].

Originally conceptualized as linked open data, the term has recently evolved to encompass requirements such as open licensing, machine-readability, non-proprietary formats, adherence to open standards, and the provision of links to related resources [5,6].

Research organizations are prolific producers of scientific digital output, generating increasingly high volumes of data within brief periods which makes robust data management practices necessary [7]. Research data management (RDM) plays a crucial role in the discovery of scientific knowledge and reuse of data by others [8].

RDM services at universities, often managed by teams dedicated to RDM service provision, involve expertise across multiple domains and stakeholders such as libraries,

IT services, legal and ethics units, and technology transfer offices with experts providing information on intellectual property rights [9].

Thereby, RDM aims to make research data accessible and more sustainable in terms of data being less likely to be lost and more likely to be usable for others, with librarians and data stewards playing a crucial role in guiding researchers towards effective and state-of-the-art data management and sharing practices based on the FAIR principles of findability, accessibility, interoperability, and reusability, which represent some but not all 'open principles' focusing on data sharing [10]. Open data, for instance, may be lacking crucial descriptive or instructive metainformation, hampering the reusability and sustainability. Moreover, the FAIR principles outline a spectrum of degrees of openness that complements the binary concept of openness vs. closedness, and allows, among others, for the necessary restrictive access to sensitive data.

Several barriers still prevent the widespread adoption of the principle of open data in its entirety. These include privacy and legal concerns (such as data sensitivity and insufficient anonymization), theoretical constraints (such as in applied or explorative research), or potential mistaken devaluation of nonconforming studies [11]. The idea of open data representing an integral part of open science has not yet been fully accepted by all scholars, as some argue that their research is limited by the fast-paced and competitive process of explorative rather than hypothesis-driven research. Thus, they fear that this actually might benefit others more than the original creators of data [12]. Institutional inertia, for example, a lack of investment in information and communication technologies, hinders efforts of making scientific processes more open to the scientific community and beyond [13]. However, these factors limiting the implementation of open data are considered to be avoidable or surmountable through measures such as strengthening inter-academic collaboration and recognizing the value of both inductive and abductive theorizing while also promoting and encouraging the preregistration of studies [14].

The open science movement has also targeted collaborations between universities and industry while involving several influencing factors facilitating open data practices with the given examples of state, corporation, profession, and sustainability-based community logic, with the latter holding the potential to overcome other barriers [15]. As such, sustainable values not only foster open science practices but also support the societal transformation towards sustainability, which, in turn, facilitates open knowledge sharing independently of the openness movement.

In general, inter-institutional collaborations, such as research communities and libraries sharing best practices on transparency, are key to fostering open science [16]. The open innovation model describes universities as essential for knowledge transfer processes towards innovative developments [17]. Knowledge transfer centers are nationwide projects meant to link universities involving different targets and sources with specific information to improve and accelerate the exchange of knowledge and expertise [18]. As one regional example for the promotion of inter-institutional collaboration in the provinces of Styria and Carinthia, the Knowledge Transfer Centre South (WTZ Süd III), coordinated by the Graz University of Technology and funded by the Austrian investment agency (AWS), encourages cooperation and exchange between academic research, business, and society. The project Open Data in Practice (ODIN) is part of the WTZ programme, and is coordinated by the University of Applied Science FH JOANNEUM Graz. The project facilitates the exchange of regional examples of institutional open data strategies that have led to the subsequently described collection and institutional guide to promoting open data practices.

This study is intended to address the question of how open data practices can be promoted in research-performing organizations in general. Which measures are already in place in exemplary regional institutions that could be adopted by others? In this process, the varying availability of resources among organizations has to be considered. Which elements can be covered to demonstrate the possible application of such measures in regard to necessary implementation resources based on time and allocation of roles? A framework will be established to improve institutional data dissemination practices. This initiative

aims to enhance data transparency and accessibility for end users, thereby increasing its value and reuse within research communities and society at large. Furthermore, it will support innovation by fostering a more cohesive and efficient distribution of information.

## 2. Literature Review

The open sharing of research data plays a crucial role in promoting research integrity by enabling results to be replicated and reproduced [19]. In the pursuit of reproducibility within experimental science, several definitions have been introduced by the International Vocabulary for Metrology adopted by the Association for Computing Machinery to facilitate clarity and coherence in scientific discourse [20,21]. *Reproducibility*, the overarching objective, demands not only repeatability but also the capability for diverse researchers to achieve consistent results amidst varying conditions, and defines the measurement as being obtainable with stated precision by a different team, a different measuring system, in a different location, and in multiple trials. In comparison, the term *Replicability* defines that the measurement can be obtained with stated precision by a different team using the same measurement procedure, the same measuring system, under the same operating conditions, in the same or a different location, and in multiple trials. And the term *Repeatability* defines that the measurement can be obtained with stated precision by the same team using the same measurement procedure, the same measuring system, under the same operating conditions, in the same location, and in multiple trials. Reproducibility differs from repeatability, which measures the variation in results under the same conditions, with the same instruments, in the same location, following the same procedure over a short period of time [22]. Open data thereby constitute a critical component for reproducibility. The open movement promoting open access, open educational resources, open source, and open science in general during the past decades is based on the idea of a collaborative culture empowering the open sharing of data [23,24]. The shift towards open science, also known as Science 2.0 or open scholarship, encompasses a variety of strategies proposed for institutions and funders. These include aligned Horizon Europe work programmes and incentives to share research outputs [25].

Academic governance is concerned with scholarship, performance, and conformance by allocation of resources and enacting policies [26]. Policies as well as guidelines determine courses of actions, and policies are based on guidelines in order to reach certain outcomes [27]. A policy has a mandatory character and a guideline is meant to be an indicative reference or recommendation. Policy guidelines, published by the United Nations Educational, Scientific and Cultural Organization (UNESCO), support knowledge-based decision-making for the adoption of open science policies, and aim to strengthen national research systems [28]. Some exemplary guidelines have already been commonly adopted by universities which detail the topic of good scientific or research practice, and academic research ethics [28–30]. Open access, a core component of open science, has been adopted as a policy by a select number of universities. However, broader open science policies and guidelines, including open data, are still not widely embraced [31]. Still, science policies as well as institutional guidelines can help to maintain, increase, and diffuse knowledge by assuring its conservation and encouraging cooperation among all branches of intellectual activity [28].

Other factors influencing university and individual researcher practices are visibility, related credibility, and reputation increasing recognition by the international community while ongoing efforts exist to reward open science practitioners and contributors [32]. Raising awareness is another strategy to promote open data practices, originating from pedagogical techniques [33]. Universally, training plays a crucial role in implementing practices for disseminating open research data, alongside principles of open science [34]. Grant funding agencies can have a significant yet contentious role in advancing open science [35].

The current international survey by Digital Science, Figshare, and Springer Nature on the state of open data argues that actions need to be taken to better support communi-

ties in adopting open data [36]. Recommendations have been co-created by researchers, research-performing institutions, and funders on strategies to overcome the barrier of the resource-intensive nature of open research, next to the development of constructive reward and recognition practices [37]. These include, amongst others, the transparency and the support of costs for open research practices, the exchange with less-resourced institutions, open research infrastructures, and collaboration between funders, research institutions, and all regional as well as global stakeholder groups towards a reformed research culture of openness and responsible research and innovation. Additionally, open science communities have been described as playing a key role in fostering the uptake of open science practices not only by a selected subset but a majority of researchers [31]. Institutions should thereby support activities by open science communities through encouraging community participation and creation, funding, and appointing local ambassadors.

While open science has been addressed many times in recent (meta)science publications and activities [25,31,37,38], this work will focus on the integral component of open data together with application-oriented strategies to foster corresponding practices at research-performing institutions.

## 3. Materials and Methods

During a workshop organized by FH JOANNEUM and Graz University of Technology (moderation and concept by the Research Data Management Team at TU Graz), representatives came together to discuss open data practices at their institutions. Participants from WTZ Süd cooperation partners, including the University of Applied Science FH JOANNEUM Graz, Graz University of Technology, the University of Graz, the University of Klagenfurt, the University of Applied Sciences Campus 02 Graz, and Graz University of Music and Performing Arts, next to a staff member from the Gustav Mahler private university for music, are detailed in Table 1. They listed examples of institutional strategies towards the commitment to open data. Non-participating cooperation partners of WTZ Süd have been the Medical University of Graz, Montanuniversität Leoben, and the University of Applied Sciences Kärnten Spittal, who are mentioned to ensure a comprehensive overview.

**Table 1.** Overview of participants and associated institutions.

| Institution | Personnel | Number of Students | Number of Participants | Participants Related Organizational Subunits |
|---|---|---|---|---|
| FH JOANNEUM | ~750 | ~5000 | 4 | Research Organization and Services, Library, Continuing Education and Study Administration |
| Graz University of Technology | ~3900 | ~13,700 | 3 | Library, Central Information Technology, Research services |
| University of Graz | ~4700 | ~30,000 | 3 | Library, Research Management and Service, IDea_Lab |
| University of Klagenfurt | ~1700 | ~12,700 | 1 | Research Services |
| Campus 02 | ~150 | ~1500 | 1 | Information Technologies and Business Informatics |
| Graz University of Music and Performing Arts | ~760 | ~2300 | 1 | Research Service |
| Gustav Mahler private University for Music | ~115 | ~260 | 1 | Research Service |

The half-day workshop in October 2023 was designed around the inquiry on strategies to promote open data practices at research-performing organizations involving selected participants from various research-supporting service units. Participants were previously directed to gather examples of open digital resources from their institutions. This was aimed at acquainting them with both the basics and the latest advances in open data practices within their respective institutions. After an introductory session on current national developments of open data and open science, there was a working group session dedicated to exchanging the collected institutional open data examples. A subsequent round-table discussion elaborated on the question of how to facilitate and foster the generation and release of open data. All examples collected in discussions were associated with topics during the phase of group work. After filtering duplicates, the examples were clustered

into categories by the first author according to their subject matter based on the associated topics selected by the participants. In this process, the list was narrowed down to 5 main content groups, summarized in Figure 1.

Furthermore, category weights were developed by the first author for the criterium of time resources allocated to an action, resulting in the scoring of categories. All examples mentioned are based on either concluded real implementations, developments, or still ongoing transformation processes from the last few years and could be associated with time courses of weeks to years. The exemplary actions were weighted by applying an arbitrary time factor between $t_w = 1–10$ which indicates the estimated necessary preparation time including the corresponding knowledge building to the action's implementation. Low factor values imply a short preparation time frame of weeks to months while a high value corresponds to one or several years depending on the resources available. Based on this time factor, examples were organized and compiled according to the guideline template.

Additionally, the individual actions were assigned to stakeholders who are involved in the implementation on the one hand, or affected by the implementation on the other hand. All associated stakeholders were listed and categorized into 5 main groups by the first author, presented in Figure 1. The defined stakeholder groups associated with the individual given examples, summarized in Table A1, were counted and summed up to depict the overall participation of stakeholders over all actions and over time-based groups of actions, summarized in Table A2.

## 4. Results

The knowledge transfer between participating institutions resulted in a list of examples that are already being or are planned to be applied at the respective organization. Its structured summary is intended to represent a starting point for implementing key steps to sustainably foster open data practices at research-performing organizations.

### 4.1. Strategies towards an Organization-Wide Commitment to Open Data

Figure 1 summarizes the collected examples on promoting open data practices among the participating institutions. Therein, all examples have been grouped according to their related subject matter. Category boxes are symbolically overlapping where content could be partly associated to another category.

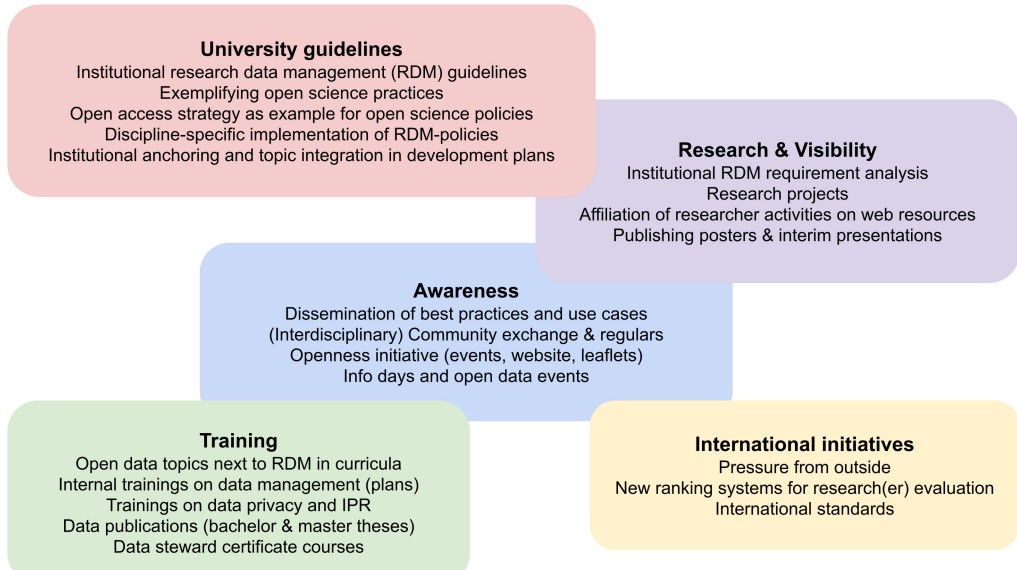

**Figure 1.** Topic-based overview of practices mentioned by workshop participants as having the potential to, or currently do, support open data.

4.1.1. University Guidelines for an Organization-Wide Commitment to Open Data

Guidelines function serve as a tool for decision-making and are intended to systematically support communities in terms of quality standards. Policies often depend on the latter and constitute a documented course of specific actions proposed by its organization. These sets of rules have to be applied by the respective target groups.

An institutional research data management (RDM) guideline exemplifying open science practices represents one way to promote open data creation. An open access strategy is a specific example for open science policies. A more sustainable approach to the application of guidelines are community-based refinements and elaborations. Such an example was given for discipline-specific implementations of RDM policies. These guidelines can take various forms, such as (well-intentioned) advice, flexible directives, binding instructions, or even lead to compliance monitoring. Therefore, guidelines can also be dynamically structured to accommodate potential changes, a necessity in the rapidly evolving fields of data economy and data management, where technological advancements and swift development are commonplace. Moreover, supervisory higher level bodies can develop guidelines and initiatives, and their progress can also be monitored both qualitatively and quantitatively. This monitoring can include knowledge scoreboards, impact assessments, as well as measures of visibility and control.

As an alternative to developing and implementing customized guidelines, internationally recognized guidelines are available that can be directly referenced or partially adopted. This approach could save time and resources. Additionally, some institutions belong to larger organizations that offer a basic set of rules, implying specific obligations.

Finally, research-performing organizations can institutionally anchor the topic of open data in development plans.

4.1.2. Research and Visibility

Open data, and especially open access, enhance the visibility of the authors involved. Moreover, there are various indirect strategies to combine the concept of open data with research activities.

First, an institution can start with an RDM requirement analysis for their researchers to identify critical issues. This helps in raising awareness about individual data practices among researchers and establishes a starting point for developing an improvement strategy that could integrate open science principles.

Second, research projects on related scientific issues or based on the reuse of open data can aid in raising awareness about the importance of the subject. Planning and funding of such activities could be integrated in project proposals. Large-scale and EU projects can develop open data strategies which can be used by individual institutions to foster open data strategies.

Additionally, promoting the institutional activities of researchers through international web resources also draws attention to related practices. One example would be a recommended or even obligatory indication of a researcher's affiliation in GitHub repositories created or edited. Publishing posters and interim presentations, including those by guest researchers, represent another method of enhancing the visibility of a research organization, alongside its scientific groups, individual scientists, and students. Such activities could be supported by media offices and science communication units. Allocating resources is another strategy to support open data initiatives. This could include basic funding for developing open data infrastructure, hiring additional personnel specialized in open data for large projects (third party-funded or provided in-kind by the institution), or offering prizes for special showcases and exceptional examples, such as open data awards.

4.1.3. Awareness Raising for Open Data

In particular, the latter category focusing on research and visibility can serve as exemplary measures in promoting open data. Additionally, there are several possible

actions to promote the concept without a specific focus on research. The following examples, while partially aligning with the latter category, aim to address a larger target audience.

The dissemination of best practices and use cases on open data is a straightforward yet effective measure to raise awareness among scientists about practical procedures and the associated topic. Participating, establishing, and coordinating community exchange—with an emphasis on interdisciplinary collaboration and regular meetings—are actions that raise awareness and contribute to knowledge creation [39]. National examples for awareness and community building are projects like FAIR Data Austria and Shared RDM [40,41]. Further initiatives focusing on knowledge transfer, such as WTZ, can establish a foundation for fostering open data practices. An openness initiative might include events, dedicated websites on specific topics, and/or distribution of leaflets. On a smaller scale, this idea can also be implemented through single information days and open data events.

### 4.1.4. Training for Open Data Practices

Comprehensive training enables a sustainable knowledge foundation. To date, the topic of open data has been of interest to a select group of scholars and has not been widely incorporated into academic lectures or curricula.

Open data can be taken up next to RDM in curricula. Internal training focused on data management and data management plans, which illustrate the practical application of open data, serve as another instrument for training staff members. Related training sessions on data privacy and intellectual property rights are examples of how staff education can be enhanced. Additionally, staff training could incorporate certificate courses for data stewards. For student training, it could be recommended or even made mandatory to publish data in the context of bachelor's and master's theses.

Additionally, there are specific funding, fellowships, awards and summer/winter schools for training and upskilling in the areas of data management and open data. Furthermore, some research programmes for early postdocs provide additional funding for trainings as well as require to include a concept for skill development. Early-stage scientists and postdocs must be systematically informed of such possibilities by, for example, research managers and similar officials.

### 4.1.5. International Initiatives for a Sustainable Implementation of Open Data Practices

International initiatives can have a top-down effect on regional communities as well as the individual institution, depending on their dimension and participating parties.

Political involvement can direct and lead to the implementation of suggested measures, such as external pressure from funding organizations, ministries, or publishers and journals. Previous examples are given by funding agencies which require their investigators to deposit publications in public libraries [42]. New infrastructures could be prioritized and financed through public and joint efforts, bringing together scientific communities for further developments and establishing rules that enable the trans- and supra-institutional participation of scientists.

Other promising measures for the future include new ranking systems to evaluate research and individual researchers. One sustainable approach to foster open data practices involves incorporating relevant principles into international standards, applying these principles, or even establishing new standards. The most prominent example is the European Open Science Cloud that aims to develop a web of FAIR data and services for science [43]. This is one example of the Common European Data Spaces that are intended to be data ecosystems for users in similar sectors [44]. Another prominent example is the OpenAIRE partnership, which aims at fostering a permanent open scholarly communication infrastructure to support European research [45]. More generally, the European Commission has proposed actions and policies that focus on generating value for economy and society through the reuse of information in research and the public sector [25].

*4.2. Guiding Template towards an Institutional Commitment to Open Data*

The exemplary actions for promoting open data at individual higher education institutions and research-performing organizations have been summarized according to their necessary preparation time together with the duration of their implementation. The results are presented in Table 2, which lists key steps of actions promoting open data practices according to their short-, medium-, or long-term applicability. This sorted list has been compiled as a step-by-step guide of possible actions applicable and amendable by interested parties in order to develop a tailored roadmap for implementing open data practice and strategies in an organization.

**Table 2.** Guideline on strategic measures for promoting open data practices within higher education institutions and research-performing organizations, with $t_w$ = time-weight factor between 1 = quick to apply, to 10 = implementation needs longer preparation time.

| Introductory Steps | $t_w$ |
|---|---|
| Participating in interdisciplinary exchange and joint event series on RDM. | 1 |
| Staff training on research data management (e.g., data steward certification). | 2 |
| **Short-Term Initiatives** | |
| Integration of the topic of open data in curricula and teaching practice. | 3 |
| Specification of good practices on data publication (bachelor and master level). | 3 |
| Affiliation of published digital objects on international web resources. | 4 |
| Training on data privacy and intellectual property rights. | 4 |
| Internal training on RDM and data management plans. | 4 |
| Survey on RDM requirements among institutional researchers. | 4 |
| Roadshows and institutional openness initiatives. | 4 |
| Institutional open science sub-website and communicating open data advantages. | 4 |
| Highlighting and disseminating exemplary use cases and best practices. | 5 |
| **Medium-Term Activities** | |
| Institutional RDM policy integrating open data. | 6 |
| Publications of posters and presentations from interim reports and guest scientists. | 6 |
| Institutional anchoring of open data matters in development plans. | 7 |
| Awareness raising in study programs, including bachelor studies. | 7 |
| Discipline-specific implementations of RDM policy. | 8 |
| Compulsory courses on RDM—integration in curricula. | 8 |
| **Long-Term Actions** | |
| Establishing and implementing international standards. | 9 |
| New ranking systems on institutional and research(er) evaluation. | 10 |
| Political involvement towards pressure from outside by, e.g., funders, publishers. | 10 |

*4.3. Institutional Stakeholders for Fostering open Data Practices*

The actions outlined in the guide have been aligned to both the parties involved in their implementation as well as the target groups impacted by these measures. These roles have been summarized in relation to the general categories presented in Figure 2.

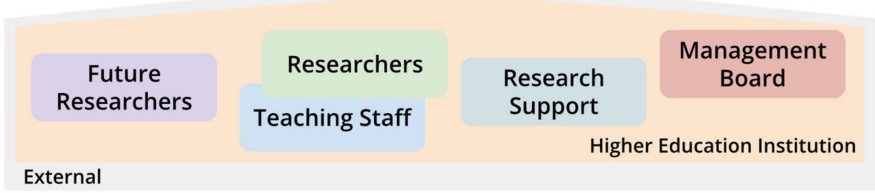

**Figure 2.** Overview of internal stakeholders as potential target groups or involved parties in strategic measures for promoting open data at higher education institutions, based on the allocation of stakeholders to actions during the analysis of collected data from workshop participants.

Future researchers are defined as students interested in pursuing a scientific career. Researchers include PhD students, senior scientists, research group leaders, as well as scientific technicians to a lesser extent. This category overlaps with that of teaching staff, as roles of researchers and teachers are frequently represented by the same individual. Research support consists of several possible service units, including IT departments, research management, research data management, technology transfer offices, legal departments, ethics committees, and similar contact points for researchers. The management board can be comprised of different (higher) hierarchy levels varying between institutions and can include rectorates, faculty, or institute heads, curricula committee members, and members from similar bodies. External stakeholders include policy makers and regional, national, or international legal bodies, funding agencies, media and publishers, or collaboration partners, that can influence internal stakeholders' fractions individually, differently, or similarly.

The summary of target groups and involved parties in the overall defined actions is visualized in Figure 3. The stakeholder groups of current and prospective researchers are mostly affected by the actions as visualized in (A). Involved parties, presented in (B), do not include all stakeholder groups that have been defined for the two matters of targeted and involved parties as future researchers are not part of institutional staff. The complete list of the corresponding stakeholder groups attributed to actions can be found in Table A1. A detailed stakeholder overview of relative time-categorized actions is presented in Figure 4. Introductory steps mainly affect and involve currently employed researchers and research support staff. Short-term initiatives further additionally incorporate future researchers and the management board into the aforementioned stakeholder groups. Medium-term activities primarily focus on researchers and future researchers, but are not limited to these groups, involving a combination from all categories. Long-term activities involve the three categories of researchers, research support, and management board, while the corresponding actions affect all stakeholder categories.

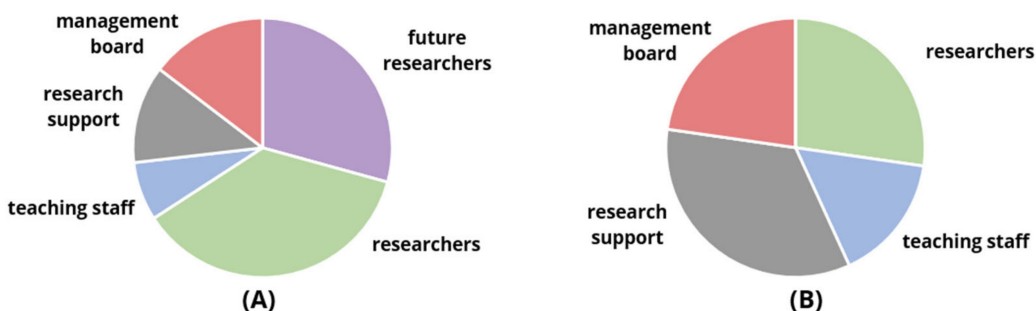

**Figure 3.** Overall internal stakeholder fractions of (**A**) target groups, and (**B**) participating parties in the implementation of measures for open data practices, based on the allocation of stakeholders to actions during the analysis of collected data from workshop participants.

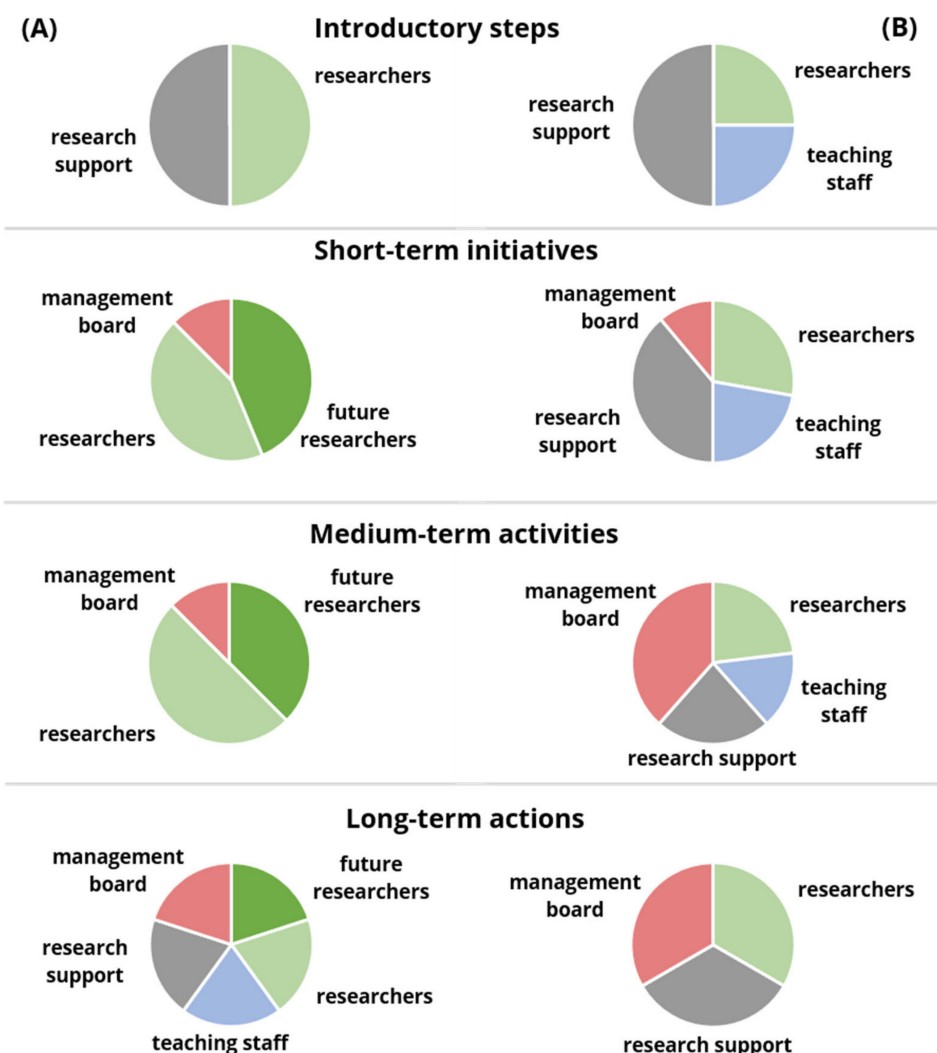

**Figure 4.** Category-based stakeholder fractions of (**A**) target groups, and (**B**) participating parties in the implementation of measures for open data practices.

## 5. Discussion

This paper compiles strategies for enhancing open data practices in research-performing organizations, drawing on insights from a workshop involving WTZ partners, detailed in Section 2. While the importance of open data is recognized by all workshop participants, the requirements and strategies differ between institutions (and also between disciplines). An exchange such as the one that took place in the context of the WTZ workshop can foster a fruitful dialogue among partners and enable them to develop personalized roadmaps for open data. Thus, the guiding template presented here aims to act as a shared framework and outcome, applicable across various institutions and organizations.

The guiding template is intended to provide a starting point for a roadmap towards an innovation playbook for fostering open data practices at academic institutions. Similar concepts for related topics have been developed for related areas, such as institutional research data management strategies [46]. The various strategies involve different actions as well as stakeholders and are categorized by the underlying umbrella topic in order to understand the conceptual representation [47]. The thematic categories give an overview of all examples by distinguishing the subject matters. Each category is a unique collection of elements and features. Still, some categories can be related to each other in different ways. For example, awareness leads to training, and training can lead to awareness. Different target groups can be addressed by both approaches. Ultimately, compulsory training programs, compared to

the other mentioned educative or awareness examples, can reach specific audiences most effectively. The two categories of research and visibility as well as guidelines could likewise be interconnected. Guidelines can support researchers in strengthening their visibility, and, indirectly, guide them on how to increase their institution's visibility. Recommendations from the literature state training to be key to implementing open research data dissemination practices and awareness raising as exemplary didactic methods [33,34]. Visibility, related credibility, and recognition have been noted to significantly impact the university and individual researcher practices, supporting responsible research and innovation. Additionally, higher level guidelines have been published [28–30,32]. Moreover, compulsory specifications on how to manage publications can comprehensively or completely reach a specific audience. Solely, the thematic category of international activities is also equivalent to a strategic category, namely the last group of examples framed as long-term actions. This category encompasses the most sustainable yet complex actions, making them challenging to implement. Consequently, it was deemed appropriate to establish a separate category during the thematic and strategic clustering process.

The strategic template is additionally presented quantitatively using arbitrary time factors that can be translated into ranges between months and years. They are a result of the variations in resource availability and allocation possibilities at individual academic institutions. Still, the applicability of the template remains unchanged and strategies can be formed on its basis. This quantitative frame can be used as a step-by step planning instrument that allows for the categorization of short- to long-term activities. Additionally, all actions have been associated with corresponding stakeholders that are either participating in their implementation or are targeted. This allows key actors to be identified and the further evaluation of necessary resources for the respective tasks. Medium-term and long-term actions require the major participation of management boards as upper management is able to function as the decision-maker for transformation processes. Long-term actions are the most sustainable way of implementing changes towards open data by default and solely affect all defined stakeholder groups. However, external stakeholders have not been included in the analysis. Consequently, this could have an effect on the above-described institutional stakeholder groups. Conducting a more detailed stakeholder analysis that considers the various external influences could be a direction for future research. Further co-creation frameworks could be made use of to extend the study beyond research-performing organizations and by involving additional stakeholders associated to research and development in general. The regional point of view could constitute another limitation of this study and could be complemented by a discussion round among international stakeholder groups. Future studies could likewise extend the underlying focus of this work on dissemination practices of open data to integrating further elements of open and responsible research and innovation.

In summary, fostering open data practices requires cultural change, including an elaborated reward system, outcome-oriented training programs, awareness-raising agents, practical application of open data in research projects and training, an infrastructure featuring wide access possibilities, and integrating various communities in a most comprehensive manner which is also in line with the goals set by the project "Facilitate Open Science Training for European Research" (FOSTER) [48]. Globally, the transformation of research assessment is essential for open data to become the new normal. This is in line with the researchers' concern of not receiving recognition for open data sharing [35]. In detail, a disciplinary approach to the global research data management support has been reported to be necessary due to variations in different subject expertise and geographies [35,48].

## 6. Conclusions

Open data have evolved over the past years to decades into a fundamental part of open science and responsible research and innovation [2,3,36,37]. Some reforms are already underway, such as larger investments in information technology infrastructures, exemplary policies emphasizing open data practices, and an overall tendency and collective need

towards the collaborative nature of sustainable science. Institutions can continue along as well as supporting this movement by implementing practical measures to promote open data practices. These involve university guidelines, research and visibility activities, awareness-raising actions, and training, next to taking part in international initiatives. Future challenges, such as artificial intelligence and the associated data privacy concerns [49], will have to be handled by collaborative communities fostering inter-institutional knowledge exchange in order to facilitate transparency, citizen engagement, as well as data-driven knowledge creation, and innovation through open data.

**Author Contributions:** Conceptualization of research and workshop, C.J.-Q. and C.S.-R.; methodology, C.J.-Q.; formal analysis, C.J.-Q.; investigation, C.S.-R., H.K.-P., B.Z. and C.J.-Q.; resources, C.S.-R. and I.H.-M.; data curation, C.J.-Q., C.S.-R., H.K.-P. and B.Z.; validation, C.J.-Q.; writing—original draft preparation, C.J.-Q.; writing—review and editing, C.J.-Q., C.S.-R., H.K.-P., B.Z., I.H.-M. and L.P.; visualization, C.J.-Q.; project administration, C.S.-R.; funding acquisition, H.K.-P. and C.S.-R. All authors have read and agreed to the published version of the manuscript.

**Funding:** The WTZ Süd is financed by AWS, by means of the National Foundation of Research, Technology and Development (Austrian Funds).

**Data Availability Statement:** All the data supporting the findings of this study are available within the manuscript, no further data are associated with the article.

**Acknowledgments:** We thank all participants from the workshop and in general all open data providers fostering open science principles.

**Conflicts of Interest:** The authors declare no conflicts of interest. The funders had no role in the design of the study; in the collection, analyses, or interpretation of data; in the writing of the manuscript; or in the decision to publish the results.

## Appendix A

**Table A1.** Overview of actions for fostering open data practices and stakeholder shares in regard to target groups and involved parties.

| Actions | Target Groups | Involved Parties |
|---|---|---|
| Interdisciplinary RDM exchange and joint event series | Research support, researchers | Research support, researchers |
| Staff training in RDM | Research support, researchers | Research support, Teaching staff |
| Open data topic integration in existing teaching | Future researchers | Teaching staff |
| Survey on RDM requirements among institutional researchers | Researchers, Management Board | Research support, Researchers |
| Specification of good practices on data publication | Future researchers | Research support, Teaching staff, Researchers |
| Affiliation of published digital objects | Researchers | Researchers, Management Board |
| Data privacy and intellectual property rights training | Researchers, Future researchers | Researcher support, Teaching staff |
| Internal training on RDM and data management plans | Researchers, Future researchers | Researcher support, Teaching staff |
| Roadshows and institutional openness initiatives | Management boards, Researchers, Future Researchers | Research support, Researchers, Management Board |

**Table A1.** *Cont.*

| Actions | Target Groups | Involved Parties |
|---|---|---|
| Institutional communicating open data advantages | Researchers, Future researchers | Research support, Researchers |
| Highlighting exemplary use cases and best practices | Researchers, Future researchers | Research support, Researchers |
| Institutional RDM policy integrating open data | Researchers | Management Board, Research support, Researchers |
| Posters, interim reports and guest scientist releases | Researchers | Research support, Researchers |
| Open data matters in development plans | Management boards, Researchers, Future Researchers | Management Board |
| Awareness raising in study programs | Future Researchers | Teaching staff, Management Board |
| Discipline-specific implementations of RDM policy | Researchers | Management Board, Research support, Researchers |
| Compulsory courses on RDM | Future Researchers | Teaching staff, Management Board |
| Establishing and implementing international standards | All | Management Board, Research support, Researchers |
| New institutional and research(er) evaluation | All | Management Board, Research support, Researchers |
| Political involvement towards pressure from outside | All | Management Board, Research support, Researchers |

**Table A2.** Summarized stakeholder shares for time-based categories, as of introductory steps (intro), short-term initiatives (short), medium-term activities (medium), and long-term actions (long), in regard to target groups and involved parties.

| Stakeholder | Overall | | Intro | | Short | | Medium | | Long | |
|---|---|---|---|---|---|---|---|---|---|---|
| | Target | Involved | Target | Involved | Target | Involved | Target | Involved | Target | Involved |
| future researchers | 13 | 0 | 0 | 0 | 7 | 0 | 3 | 0 | 3 | 0 |
| researchers | 16 | 12 | 2 | 1 | 7 | 5 | 4 | 3 | 3 | 3 |
| teaching staff | 3 | 7 | 0 | 1 | 0 | 4 | 0 | 2 | 3 | 0 |
| research support | 5 | 15 | 2 | 2 | 0 | 7 | 0 | 3 | 3 | 3 |
| management board | 6 | 10 | 0 | 0 | 2 | 2 | 1 | 5 | 3 | 3 |

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
