# Peer review of "Fostering Open Data Practices in Research-Performing Organizations"

_publications, doi:10.3390/publications12020015_

Round 1

Reviewer 1 Report

Comments and Suggestions for Authors

I have suggested two comments (not mandatory) - insert into the file - concerning some definitions. I refer to reproducibility concern. I suggest to insert into the paper these two brief content

Reproducibility differs from repeatability, which measures the variation of results under the same conditions, with the same instruments, in the same location, following the same procedure over a short period of time. [Jim Al-Khalili, The Joy of the science, 2022]

In the quest for reproducibility in experimental science, the absence of uniform terminology has caused confusion among research communities. To address this issue, we offer clear definitions inspired by the International Vocabulary for Metrology (VIM). Metrology is the scientific study of measurement, it establishes a common understanding of units, crucial in linking human activities

Repeatability, grounded in metrology's core principles, signifies the anticipation of consistent outcomes within tightly controlled experiments. Reproducibility, the overarching objective, demands not only repeatability but also the capability for diverse researchers to achieve consistent results amidst varying conditions.

Drawing upon these definitions, the Association for Computing Machinery (ACM 2016)[1] has adopted standardized terminology to facilitate clarity and coherence in scientific discourse that we will use in this article.

Repeatability (Same team, same experimental setup): The measurement can be obtained with stated precision by the same team using the same measurement procedure, the same measuring system, under the same operating conditions, in the same location on multiple trials. For computational experiments, this means that a researcher can reliably repeat her own computation.

Replicability (Different team, same experimental setup): The measurement can be obtained with stated precision by a different team using the same measurement procedure, the same measuring system, under the same operating conditions, in the same or a different location on multiple trials. For computational experiments, this means that an independent group can obtain the same result using the author's own artifacts.

Reproducibility (Different team, different experimental setup): The measurement can be obtained with stated precision by a different team, a different measuring system, in a different location on multiple trials. For computational experiments, this means that an independent group can obtain the same result using artifacts which they develop completely independently.

[1] Association for Computing Machinery (2016). Artifact Review and Badging. Available online at: https://www.acm.org/publications/policies/artifact-review-badging

Author Response

Dear reviewer, thank you for your time reviewing our manuscript, we included your suggestions in the literature survey of the newly structure manuscript. All changes are highlighted in red.

Reviewer 2 Report

Comments and Suggestions for Authors

This paper presented a strategic guide for promoting open data practices from an organizational perspective based on the data collected from a workshop discussion. The topic and the presented strategies are interesting and timely. However, there are three main issues with the paper:

1.     For the Introduction section, the authors failed to provide clear research purposes and questions for the study.

2.     The paper didn’t include a literature review section to provide a theoretical foundation to support the design of this study. Some contents in the Introduction section can be moved to the literature review section.

3.     The most important issue is that the methodology section is too simple. More details need to be provided for the following questions:

-       How many participants have joined the workshop to share ideas?

-       What is the rationale for grouping the strategy samples and individual actions? Were they grouped by one author or several authors? If by several authors, did they test the agreement between them?

-       How did the first author create the category weights? Who provided the estimated necessary preparation time?

-       How did the authors process and analyze the collected data from the workshop? Did they do any coding? How did they synthesize the topics in Figure 1 from their research data?

The connection between the methodology and the results is weak at the moment.

Comments on the Quality of English Language

The writing of the whole paper is fluent and easy to follow.

Author Response

Thank you for your time spent in reviewing our manuscript and listing revising instructions. We have incorporated your recommendations, and all changes in the text are highlighted in red. We also had the complete manuscript revised by an experienced English-speaking editor.
1. We restructured the manuscript and changed the introduction, also elaborating on research questions.
2. We added a literature review section apart from the introduction as suggested.
3. We extended the methods section in terms of comprehensiveness as well as comprehensibility, and in order to improve the connection between methodology and results. We added descriptions as well as tables on data collection and analysis, grouping of actions, weights, and stakeholders.

Reviewer 3 Report

Comments and Suggestions for Authors

The article presents the categorization and results of participants' involvement in a workshop. It provides relevant results and information, as perceptions of open data actions and policies are crucial for improving proposals in research institutions. Despite the interest, I believe some areas need improvement:

Title: It needs to be more specific and reflect the specificity of the proposal presented.

Introduction: It should be better contextualized; examples are sometimes not provided, and things are cited very briefly—for example, paragraph 4, page 2. The text could invest more time in better illustrating the proposal and including guidelines for best practices in a somewhat more developed manner.

Objective: This is a problem in the text. The title is general, and the text's aim needs to be developed in the introduction or methodology. Also, research questions are not presented; this would greatly help the discussion better understand what has been obtained from the research and better interrelate the results.

Methodology: Once again, it is very concise. The participants need to be described in detail. The methodology requires further development and should be replicable to a greater extent. The issue of stakeholders needs to be clarified.

Discussion: It is suggested that conclusions be added to this section. It does not give a sense of finalization. Additionally, organizing it around research questions would help present the actual value of the proposal. Furthermore, it is suggested to include limitations.

Author Response

Dear reviewer, thank you for reviewing our manuscript and all suggested instructions as well as recommendations.
We now have changed the title.
We have restructured the manuscript and added several paragraphs to better introduce the topic. While another reviewer suggested to shorten the introduction, we have added a section on reviewed literature which was also suggested by a different reviewer. Research questions have been integrated. And a conclusion complements and summarizes the discussion of results.
We have also expanded the methods section in regard to comprehensiveness and comprehensibility. Among other changes, we added additional descriptions and a table on participants, as we have addressed the methodology on stakeholders in more detail.
All changes are highlighted in red.

Reviewer 4 Report

Comments and Suggestions for Authors

This is an interesting paper with empirical evidence on the development of open science in the field. However, some minor revision would improve its quality.

Title: All nouns, pronouns, verbs, adjectives, and adverbs should be capitalized.

Introduction: The section is a little bit too long; on the other hand, objectives are missing. You should clearly state the goals of this study in the introduction, and you should delete the last three paragraphs (lines 81 - 104) from the introduction (perhaps, you can use them in the discussion).

Materials and methods: It would be helpful to add a table with the involved organisations and the people (at least, their number). Perhap also a table with the meetings (dates...).

Results: Are the results represented in figures 3 and 4 statistically significant?

Discussion: The results should be discussed against other studies (you can make use of the last three paragraphs from the introduction, for instance) so that the reader can better understand the particularities and the interest and relevance of your findings. Also, at the end, you should add some perspectives and suggestions for further research.

Author Response

Dear reviewer, thank you for your time spent in reviewing our work and formulating the instructions as well as recommendations to improve the quality of our manuscript!
We have revised the complete document and highlighted all changes in red.
1. We have changed the title accordingly.
2. We have shortened the Introduction and added an own section on reviewed literature as suggested by another reviewer.
3. We have extended descriptions of methods and added the table of involved organisations and participants. Regarding the issue of significance: the focus of this study was set on qualitative and not on quantitative results from the workshop. This implicates a quantitative approximation, for presentation purposes only, and a low number of examples in some categories, not intended for any statistical analysis.
4. The Discussion was extended, and a conclusion added as suggested by another reviewer.

Round 2

Reviewer 3 Report

Comments and Suggestions for Authors

All the changes included satisfied the previous report. I consider the text ready to be accepted.